# A Meta-Analysis Comparing the Efficacy and Safety of Peramivir with Other Neuraminidase Inhibitors for Influenza Treatment

**DOI:** 10.3390/medicina56020063

**Published:** 2020-02-05

**Authors:** Jui-Yi Chen, Shih-Kai Wei, Chih-Cheng Lai, Teng-Song Weng, Hsin-Hua Wang

**Affiliations:** 1Division of Nephrology, Department of Internal Medicine, Chi Mei Medical Center, Tainan 71004, Taiwan; a50601@mail.chimei.org.tw; 2Department of Pharmacy, Chi Mei Medical Center, Liouying, Tainan 73657, Taiwan; walker914914@hotmail.com; 3Department of Internal Medicine, Kaohsiung Veterans General Hospital, Tainan Branch, Tainan 71051, Taiwan; dtmed141@gmail.com; 4Department of Pediatrics, Chi Mei Medical Center, Liouying, Tainan 73657, Taiwan

**Keywords:** peramivir, neuraminidase inhibitors, influenza

## Abstract

*Background and Objectives:* This meta-analysis compared the efficacy and safety of peramivir compared to other neuraminidase inhibitors (NAIs). *Materials and Methods:* Data from PubMed, Embase, and Cochrane databases and ClinicalTrials.gov were searched until January 2019. Randomized controlled trials (RCTs) and observational studies (OSs) comparing peramivir with other NAIs for treating influenza were included. The Grading of Recommendations, Assessments, Development, and Evaluations (GRADE) system was used to judge the overall certainty of evidence; the result was moderate. The primary outcome was time to alleviation of symptoms. Twelve articles involving 2681 patients were included in this meta-analysis. We used a random-effect model to pool the effect size, which is expressed as the difference in means (MD), risk ratio (RR), and 95% confidence interval (CI). *Results:* Overall, peramivir was superior to other NAIs (MD = −11.214 hours, 95% CI: −19.119 to −3.310). The incidence of adverse events (RR = 1.023, 95% CI: 0.717 to 1.460) and serious adverse events (RR = 1.068, 95% CI: 0.702 to 1.625) in the peramivir group was similar to those in the oseltamivir group. In addition, peramivir had higher efficacy than each NAI alone. *Conclusion:* In conclusion, the efficacy of peramivir might be higher than that of other NAIs, and this agent is tolerated as well as other NAIs.

## 1. Introduction

Influenza is one of the most widespread infectious diseases worldwide. Despite vaccination, influenza affects nearly everyone at some point in their life and sometimes leads to serious syndromes. Thus, relevant drugs are important in medical care. Three classes of drugs, namely adamantanes, neuraminidase inhibitors (NAIs), and selective inhibitors of influenza cap-dependent endonuclease, are currently used to treat influenza [1,2]. However, adamantanes can only be used to treat influenza A (H1N1 and H3N2), and influenza viruses have developed more resistance to adamantanes [3]. Although influenza cap-dependent endonuclease is a new oral drug, it is only approved for patients older than 12 years, and studies with head-to-head comparisons are limited [4]. Therefore, NAIs have become vital for influenza treatment.

Currently, four NAIs, namely oseltamivir, zanamivir, peramivir, and laninamivir, are available. Among them, peramivir is the only NAI administered intravenously, and it has been approved to treat nearly all patients, unless injection is contraindicated. Therefore, peramivir could be a useful option of treatment for most patients with severe symptoms [1,2]. However, recent meta-analysis [5] comparing peramivir to oseltamivir only determined the efficacy on an adult group.

Studies comparing these four NAIs for influenza treatment are limited. This meta-analysis compared the clinical efficacy of peramivir with the three other NAIs with respect to time to alleviation of symptoms or defervescence on groups patients older than 18 years and younger than or equal to 18 years. In addition, we compared the safety of peramivir with other NAIs by examining the risks of adverse events (AEs) and serious adverse events (SAEs).

## 2. Materials and Methods

### 2.1. Study Search Strategy and Selection

Relevant studies written in all languages were obtained through a systematic search of the literature on the PubMed, Embase, and Cochrane databases and ClinicalTrials.gov until January 2019 (Appendix A), and the following search terms were used: “peramivir or rapiacta or BCX-1812 or RWJ 270201,” “oseltamivir or tamiflu,” “zanamivir or relenza,” “laninamivir or inavir,” “neuramidase inhibitor,” and “influenza or seasonal influenza or flu or H1N1.” We included randomized controlled trials (RCTs) and observational studies (OSs), and we excluded case reports and case series. Only articles with patients that were rapid test positive and that compared intravenous peramivir with at least one other NAI were included. Studies focusing on the effects of NAIs on cell lines or animals or those conducting pharmacokinetic–pharmacodynamic assessment for NAIs were excluded. Two reviewers (T.-S.W. and S.-K.W.) searched and examined all articles to avoid bias. When they disagreed on the inclusion of an article, a third author (C.-C.L.) judged the inclusion of the article.

### 2.2. Data Extraction and Outcome Assessment

The following data were extracted from the studies: the first author name, year of publication, sample size, subtype of influenza investigated, patient inclusion criteria, patient ages, details of the treatment protocol, clinical outcomes, and AEs. The primary outcome was time to alleviation of symptoms. The secondary outcome was the incidence of AEs. All outcomes and clinical data were extracted from the articles by two reviewers (T.-S.W. and S.-K.W.). When the data were not available for a study, we tried to contact the authors to request study data.

### 2.3. Data Analysis and Study Quality Assessment

In this meta-analysis, the effect size, which is expressed as the difference in means (MD), was pooled using a random-effect model to analyze time to alleviation of symptoms (hours) in individual studies. In this analysis, a negative effect size value indicated that peramivir is a more favorable treatment option. The effect size of the risk ratio (RR) was pooled using a random-effect model to analyze the risk of an adverse event. We used Comprehensive Meta-Analysis software, version 3.3070 (Biostat, Englewood, NJ, USA) to perform statistical analysis. Heterogeneity was investigated using *I^2^* tests; *I^2^* values more than 50% indicated high heterogeneity. In addition, we used funnel plots and Egger’s test to detect the presence of publication bias. Statistics were considered significant when *p* < 0.05. Moreover, we conducted subgroup analyses for various antiviral treatments, age groups, and study design. This meta-analysis was conducted in accordance with Preferred Reporting Items for Systematic Reviews and Meta-Analyses. The quality of the included RCTs and OSs was evaluated using the Cochrane risk-of-bias assessment tool 2.0 (RoB 2.0) [6] and ROBINS-I tool [7]. The overall quality of each outcome was evaluated by the Grading of Recommendations Assessment Development and Evaluation (GRADE) system. Two reviewers (T.-S.W. and C.-C.L.) evaluated the quality of all articles to avoid bias. When they disagreed on the quality of an article, a third author (S.-K.W.) judged the inclusion of the article.

## 3. Results

### 3.1. Study Search Outcomes and Included Patients

Our initial search yielded 1183 articles, of which 278, 884, and 21 were from the PubMed, Embase, Clinicaltrials.gov and Cochrane databases, respectively. A total of 226 articles were excluded because of duplication; therefore, the titles and abstracts of 957 articles were screened. Subsequently, 26 articles were assessed for eligibility. Nine articles were excluded because they were review articles [5,8,9,10,11,12,13,14,15] and 3 articles were excluded because they did not compare peramivir with other NAIs [16,17,18]. Furthermore, one study was excluded because of crossover treatment [19] and the other one was excluded because of lack of data [20]. Finally, a total of 12 articles with complete data were selected for this meta-analysis (Figure 1). The number of patients included in each study ranged from 32 to 1091, and patient age ranged from 1.8 to 77.6 years. All articles compared peramivir with at least one NAI. Five trials [21,22,23,24,25] compared peramivir with oseltamivir only, two trials [26,27] compared peramivir with oseltamivir and laninamivir, and five trials [28,29,30,31,32] compared peramivir with oseltamivir, laninamivir, and zanamivir. The risk of bias in most studies was low (Table 1 and Table 2) and the quality of most outcomes was moderate (Table 3). Patient characteristics, patient inclusion criteria, treatment protocols, and outcomes of each study are listed in Table 4 and Table 5. Of the five RCTs and seven OSs, nine articles examined influenza A and B, two articles investigated influenza A, and one article did not mention the influenza virus type.

### 3.2. Meta-Analysis of Clinical Efficacy

Comparing the peramivir and nonperamivir groups, the overall MD of time to alleviation of symptoms was −11.214 h (95% confidence interval (CI): −19.119 to −3.310, *p* = 0.005; Figure 2). We conducted subgroup analyses in which the included studies were separated into RCTs and OSs. Peramivir had higher efficacy for time to alleviation of symptoms (MD = −14.036 h, 95% CI: −23.126 to −4.945, *p* = 0.002) in pooled analysis of OSs but not in pooled analysis of RCTs (MD = −6.758 h, 95% CI: −20.458 to 6.941, *p* = 0.334; Figure 3). In addition, the peramivir group exhibited significantly shorter time to alleviation of symptoms than the oseltamivir group MD = −11.338 h, 95% CI: −19.475 to −3.200, *p* = 0.006, Figure 4; peramivir vs. zanamivir: MD = −20.846 h, 95% CI: −31.333 to −10.359, *p* < 0.05, Figure 5; peramivir vs. laninamivir: MD = −21.571 h, 95% CI: −29.656 to − 13.486, *p* < 0.05, Figure 6. We then determined the efficacy of NAIs for various age groups. The less than or equal to 18 years group exhibited a significant favor towards peramivir (MD = −12.809 h, 95% CI: −23.396 to −2.222, *p* = 0.018), and the more than 18 years group exhibited favor towards peramivir, but without statistical significance (MD = −5.630 h, 95% CI: −13.573 to 2.314, *p* = 0.165; Figure 7).

### 3.3. Risk of Adverse Event

We assessed the risks of overall AE and SAE. The risk of AE in the peramivir group was similar to that in the oseltamivir group (risk ratio (RR) = 1.023, 95% CI: 0.717 to 1.460, *p* = 0.900; Figure 8). The risk of SAE was similar between the peramivir and oseltamivir groups (RR = 1.068, 95% CI: 0.702 to 1.625, *p* = 0.759; Figure 9).

### 3.4. Heterogeneity and Publication Bias

Overall heterogeneity, according to MD’s I^2^ tests, was 46.1%, and heterogeneity based on subgroup analysis of RCT and OS study designs was 33.2% and 44.3%, respectively. Comparing peramivir with oseltamivir, zanamivir, and laninamivir, heterogeneity was 50.6%, 4.2%, and 0%, respectively. The overall publication bias, as assessed using funnel plots (Figure 10) and Egger’s test (*p* = 0.248), was not significant. Next, we used sensitivity analysis to investigate the high heterogeneity. After excluding two studies [26,31], the heterogeneity of comparing peramivir with oseltamivir through I^2^ tests was 0%. The RR of heterogeneity for AEs and SAEs, as determined through I^2^ tests, was 47.4% and 0%, respectively. The RR of publication bias, assessed through Egger’s test, was not significant between AEs and SAEs (*p* = 0.891 and *p* = 0.609, respectively).

## 4. Discussion

This meta-analysis included five RCTs and seven OSs that involved a total of 2681 patients. The results indicated that peramivir reduced the time to the first alleviation of symptoms; thus, the efficacy of peramivir might be superior to that of other NAIs. However, subgroup analysis of study designs determined the higher efficacy of peramivir than other NAIs with respect to time to alleviation of symptoms only applied to OSs, but not RCTs. Therefore, our findings suggest that peramivir should have at least similar or even higher efficacy than other NAIs. In addition, we compared the efficacy of peramivir with each NAI, including oseltamivir, zanamivir, and laninamivir. Pooled analysis indicated that the efficacy of peramivir was higher than that of the other NAIs. This result is consistent with that of a recent meta-analysis by Lee et al. that included two RCTs and five OSs that compared the efficacy of intravenous peramivir with that of oral oseltamivir for influenza treatment [5]. That study found that peramivir might reduce time to alleviation of fever more effectively than oral oseltamivir. This meta-analysis compared the efficacy of peramivir with that of other NAIs in various age groups and revealed that peramivir achieved significantly higher efficacy than other NAIs in the less than or equal to 18 years group, but no significant difference was evident in the more than 18 years group. Lee et al. [5] discerned no significant difference in efficacy between peramivir and oseltamivir in adults, which is consistent with our finding. However, they could not perform child-specific analysis because they included only one study involving children. In summary, our finding suggests that the efficacy of peramivir is superior to that of other NAIs. This meta-analysis showed peramivir reduced the time to alleviation of symptoms by 12–24 h compared to other NAIs, these data potentially mean that peramivir can shorten hospitalization stays, decrease medical costs, avoid complications, and return patients to normal life quickly. However, this finding should be confirmed in additional studies.

In this meta-analysis, we evaluated the safety of peramivir compared with oseltamivir. Pooled analysis of four RCTs [21,22,23,28] and three OSs [24,25,29] indicated that the incidence rate of drug-related AEs and SAEs exhibited no significant difference between peramivir and oseltamivir. These findings are consistent with the results of a previous meta-analysis [5] in which peramivir was found to be as safe as oseltamivir.

This study has some differences from Lee et al. [5]. First, we included all available NAIs. Second, compared with the previous meta-analysis [5], we included more studies to analyze the efficacy and safety of peramivir. Third, we performed subgroup analysis for the more than 18 years group and less than or equal to 18 years group.

This meta-analysis has several limitations. First, the influenza virus subtype was associated with drug resistance in the included articles but not in all RCTs; therefore, we could not completely randomize virus subtypes because the method for choosing NAIs is different in each study [1] and due to patients’ preference in OSs. Second, antipyretics and analgesics were administered in some articles [21,22,23,28,30,31]. Although patients were requested to measure body temperature and symptoms at least 4 hours later in all articles, outpatients might not follow this principle. Third, we did not investigate the dosage and duration of NAI treatment in this meta-analysis. Due to inconsistency of study design among these studies, we did not investigate the dosage or the duration of NAI treatment in this meta-analysis as they hindered carrying out subgroup analysis of elderly and non-elderly adults and influenza subtype as well. Fourth, this study did not focus on severe patients because studies investigating severe patients on influenza treatment by NAIs are limited. Finally, high heterogeneity was observed in time to alleviation of all symptoms in our meta-analysis; thus, we performed sensitivity analysis. The heterogeneity significantly reduced after excluding two studies [26,31], and the result still favored the efficacy of peramivir. Yoshino applied ≤37 °C as the threshold of fever resolution; however, most studies used thresholds ranging from ≤37.5 °C to ≤38 °C. Clarifying this difference might be difficult because our included studies did not report the method of measurement. However, no evidence supports the contention that the threshold of fever resolution is associated with the treatment effect.

## 5. Conclusions

In conclusion, the efficacy of peramivir might be higher than that of other NAIs, and peramivir exhibited a similar safety profile. However, further study should be conducted to confirm this result.

## Figures and Tables

**Figure 1 medicina-56-00063-f001:**
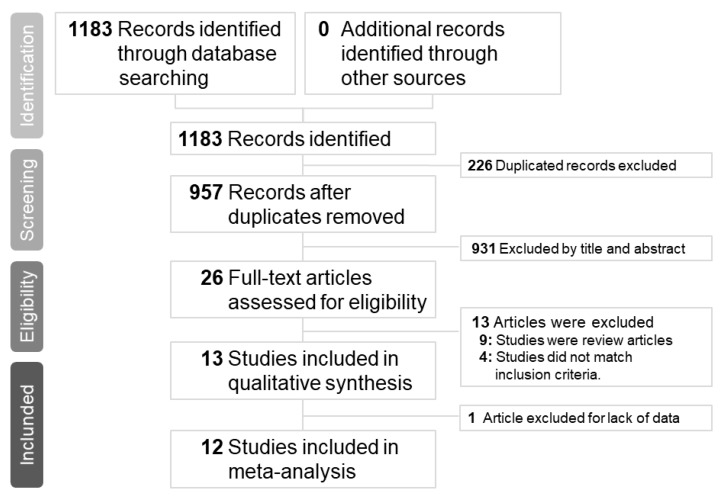
Preferred reporting items for systematic reviews and meta-analyses flow chart of the study selection process.

**Figure 2 medicina-56-00063-f002:**
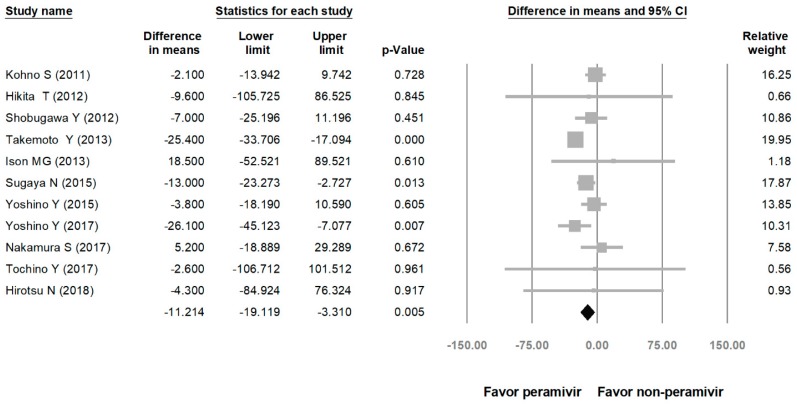
Time to alleviation of symptoms of peramivir compared with other neuraminidase inhibitors: Peramivir vs. all other NAIs. (hours). Black shapes indicate overall summary.

**Figure 3 medicina-56-00063-f003:**
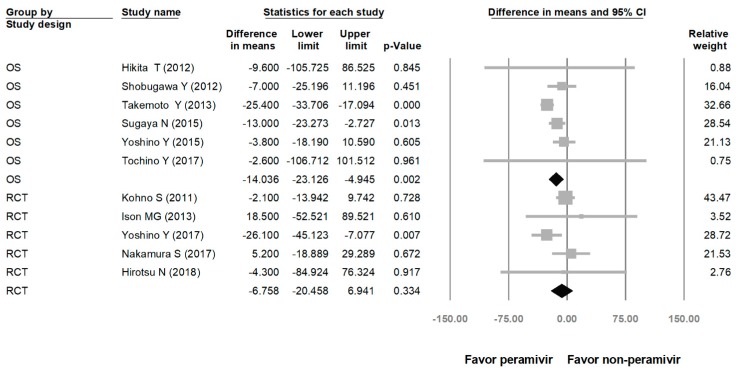
Time to alleviation of symptoms of peramivir compared with other neuraminidase inhibitors: subgroup analysis by study design (hours). OS: observational study; RCT: randomized controlled trial; black shapes indicate subgroup summary.

**Figure 4 medicina-56-00063-f004:**
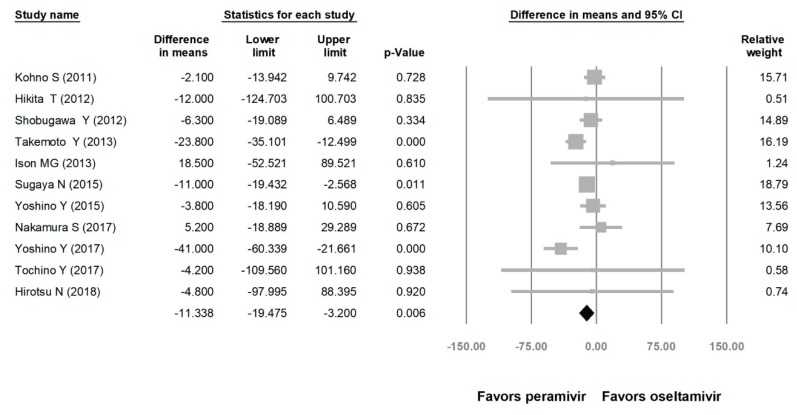
Time to alleviation of symptoms of peramivir vs. oseltamivir. (hours). Black shapes indicate overall summary.

**Figure 5 medicina-56-00063-f005:**
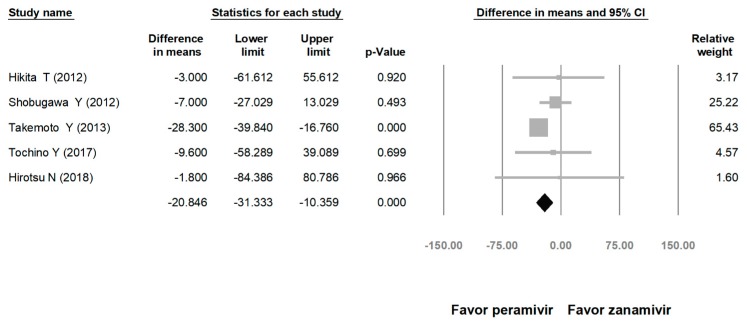
Time to alleviation of symptoms of peramivir vs. zanamivir. (hours). Black shapes indicate overall summary.

**Figure 6 medicina-56-00063-f006:**
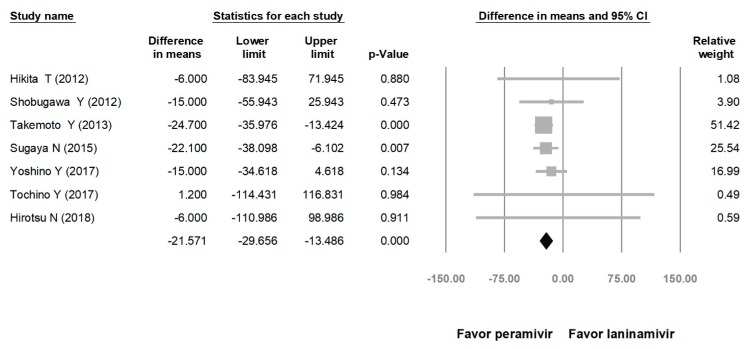
Time to alleviation of symptoms of peramivir vs. laninamivir. (hours). Black shapes indicate overall summary.

**Figure 7 medicina-56-00063-f007:**
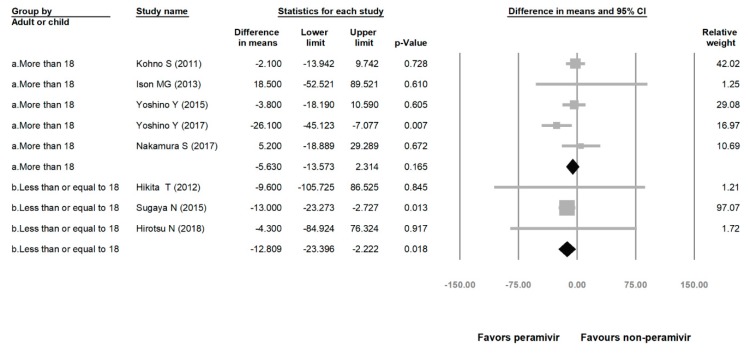
Time to alleviation of symptoms of peramivir compared with other neuraminidase inhibitors: More than 18 years group and less than or equal to 18 years group. (hours). Black shapes indicate subgroup summary.

**Figure 8 medicina-56-00063-f008:**
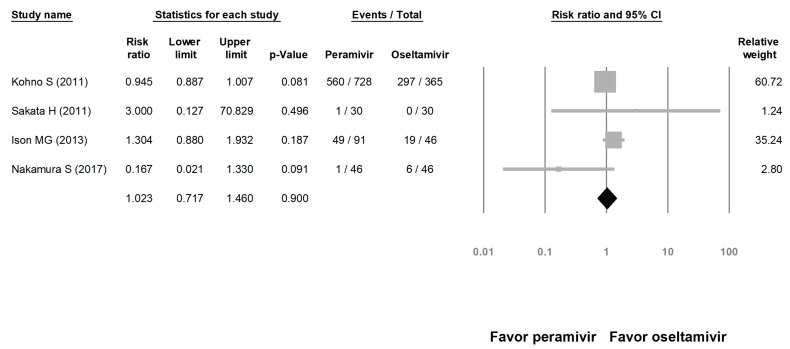
Risk of adverse events for peramivir compared with oseltamivir: overall risk of adverse event. Black shapes indicate overall summary.

**Figure 9 medicina-56-00063-f009:**
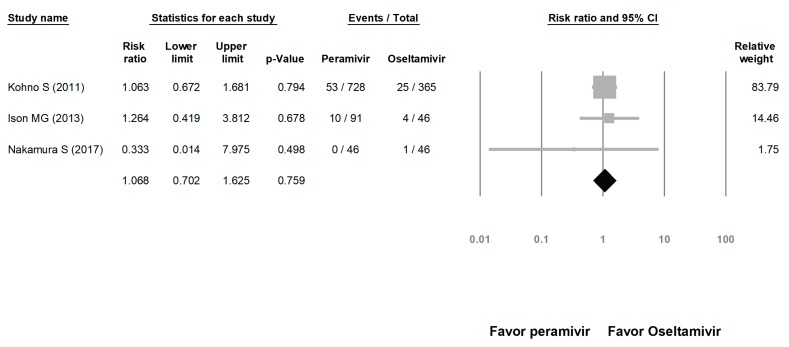
Risk of adverse events for peramivir compared with oseltamivir: overall risk of serious adverse event. Black shapes indicate overall summary.

**Figure 10 medicina-56-00063-f010:**
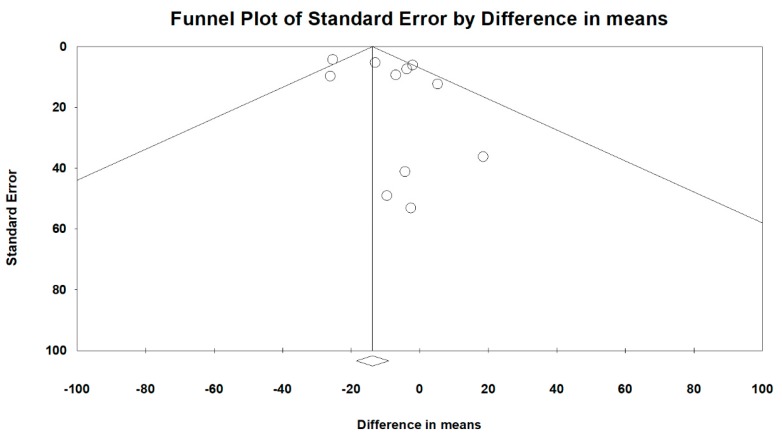
Funnel plots for the overall clinical efficacy of the included studies.

**Table 1 medicina-56-00063-t001:** Summary of risk-of-bias assessment in the meta-analysis (RoB 2.0).

Randomized (RoB 2.0)	Domains 1	Domains 2	Domains 3	Domains 4	Domains 5	Overall Bias
Kohno, S 2010	Low	Low	Low	Low	Low	Low
Ison, MG 2013	Low	Low	Low	Low	Low	Low
Nakamura, S 2017	Low	Low	Some concerns *	Low	Some concerns +	Some concerns
Yoshino, Y 2017	Some concerns+	High *	Low	High*	Some concerns #	High
Hirotsu, N 2018	Low	High #	Some concerns *	Low	Some concerns *	High

Note: # The predicted direction of bias tends toward null, * The direction of bias is unpredictable, + The predicted direction of bias favors experimental. Domains 1: Bias arising from the randomization process. Domains 2: Bias due to deviations from intended interventions. Domains 3: Bias due to missing outcome data. Domains 4: Bias in measurement of the outcome. Domains 5: Bias in selection of the reported result.

**Table 2 medicina-56-00063-t002:** Summary of risk-of-bias assessment in the meta-analysis. (ROBINS-I).

Non-Randomized (ROBINS-I)	Domain 1	Domain 2	Domain 3	Domain 4	Domain 5	Domain 6	Domain 7	Overall
Sakata, H 2011	Low	Low	Low	Low	Low	Low	Low	Low
Hikita, T 2012	Low	Low	Low	Low	Low	Low	Low	Low
Shobugawa, Y 2012	Low	Low	Low	Low	No information	Low	Low	Low
Takemoto, Y 2013	Low	Low	Low	Low	Low	Low	Low	Low
Sugaya, N 2015	Low	Low	Low	Moderate *	No information	Low	Low	Moderate *
Yoshino, Y 2015	Low	Low	Low	No information	Low	Low	Low	Low
Tochino, Y 2017	Low	Low	Low	No information	Serious **	Low	Low	Serious **

Note: * The choice of neuraminidase inhibitor (NAI) was based on patients’ or parents’ wishes, except in the group treated with peramivir. ** Of the 863 postcards that were delivered, only 263 were returned and of those only 10 patients were treated with peramivir. Domains 1: Bias due to confounding. Domains 2: Bias in selection of participants into the study. Domains 3: Bias in classification of interventions. Domains 4: Bias due to deviations from intended interventions. Domains 5: Bias due to missing data. Domains 6: Bias in measurement of outcomes. Domains 7: Bias in selection of the reported result.

**Table 3 medicina-56-00063-t003:** Summary of Grading of Recommendations, Assessments, Development, and Evaluations (GRADE) assessment in the meta-analysis.

GRADE Assessment							
Certainty Assessment							Certainty
No of Studies	Study Design	Risk of Bias	Inconsistency	Indirectness	Imprecision	Other Considerations	
**Time to Alleviation of Symptoms**							
5	randomized trials	serious ^a^	not serious	not serious	not serious	none	⨁⨁⨁◯ MODERATE
7	observational studies	serious ^b^	serious ^b^	not serious	very serious ^c^	strong associationall plausible residual confounding would suggest spurious effect, while no effect was observed	⨁⨁◯◯ LOW
**Risk of Adverse Events**							
3	randomized trials	serious ^a^	not serious	not serious	not serious	none	⨁⨁⨁◯ MODERATE
1	observational studies	not serious	not serious	not serious	not serious	none	⨁⨁⨁⨁ HIGH
**Risk of Serious Adverse Event**							
3	randomized trials	serious ^a^	not serious	not serious	not serious	none	⨁⨁⨁◯ MODERATE

CI: Confidence interval. Explanations: ^a^—problems concerning blinding; ^b^—loss to follow up in one study; ^c^—small sample size.

**Table 4 medicina-56-00063-t004:** Summary of neuraminidase inhibitors used for influenza treatment in randomized controlled trials in the meta-analysis.

Author, Year	Influenza Virus Subtype	Patient Enrollment Criteria	Sample Number (Men/Women)	Age (Mean ± SD)	Age Group	Blind	Treatment Protocol	Outcome Measurement
**Randomized controlled trials**								
**Kohno S et al., 2011**	A(H1) A(H3) B	Rapid test positive, body temperature ≧38.0 °C, two moderate to severe symptoms among seven symptoms: headache, muscle or joint pain, feverishness or chills, fatigue, cough, sore throat, and nasal stuffiness.	Peramivir: 726 (378/348) Oseltamivir: 365 (184/181)	Peramivir: 35.4 ± 11.6 Oseltamivir: 34.6 ± 11.7	>18	Yes, but no details of blind description	Intravenous with peramivir 300 or 600 mg once daily for 5 days or oral oseltamivir 75 mg twice daily for 5 days.	Time to alleviation of symptoms, change from the influenza virus titer, adverse events.
**Ison MG et al., 2013**	A(H1N1) A(H3N2) B	Rapid test positive, influenza-like illness within the previous 72 h with documented fever or feverishness, ≥1 respiratory symptom (cough, sore throat or nasal congestion), ≥1 constitutional symptom (headache, myalgia, feverishness or malaise/fatigue).	Peramivir: 81 (38/43) Oseltamivir: 41 (19/22)	Peramivir: 58.0 ± 23.0 Oseltamivir: 62.2 ± 21.1	>18	Participant, care provider, investigator, outcomes assessor.	Intravenous with peramivir 200 or 400 mg once daily for 5 days or oral oseltamivir 75 mg twice daily for 5 days.	Time to clinical stability, time to alleviation of symptoms, time to hospital discharge, time to resumption of usual activities, change from the influenza virus titer, adverse events.
**Nakamura S et al., 2017**	A(H3N2), A(H1N1) pdm09 B	Rapid test positive, body temperature ≥ 38.0 °C, treatment within 48 hours influenza illness as indicated by at least 1 symptom: headache, muscle or joint pain, feverishness or chills, and fatigue as general symptoms, and cough, sore throat, and nasal stuffiness as respiratory symptoms.	Peramivir: 46 (21/25) Oseltamivir: 46 (22/24)	Peramivir: 72.2 ± 14.1 Oseltamivir: 70.1 ± 11.1	>18	NO	Intravenous with peramivir 600 mg once daily (a second infusion at >2 days later, if necessary, was permitted) or oral oseltamivir 75 mg twice daily for 5 days.	Time to alleviation fever, time to alleviation of symptoms, change from the influenza virus titer, adverse events.
**Yoshino Y et al., 2017**	A B	Rapid test positive, axillary temperature ≥37.0 °C, influenza-like illness, including fever, muscle pain, chills, sweating, headache, dry cough, fatigue, nasal congestion, and respiratory failure.	Peramivir: 13 (6/7) Oseltamivir: 9 (3/6) Laninamivir: 12 (3/9)	Peramivir: 43.6 ± 15.3 Oseltamivir: 40.4 ± 9.84 Laninamivir: 36.2 ± 10.0	>18	NO	Intravenous with peramivir 300 mg once daily or oseltamivir 75 mg twice daily for 5 days or inhaled laninamivir 40 mg once daily.	Time to alleviation fever, alleviation of other symptoms.
**Hirotsu N et al., 2018**	A(H1N1) A(H3N2) A(H1N1) pdm09 B	Rapid test positive, axillary temperature ≥37.5 °C, influenza-like illness (cough, sore throat, headache, nasal discharge, muscle or joint pain, and fatigue).	Peramivir: 28 (15/13) Oseltamivir: 30 (22/8) Zanamivir: 26 (9/17) Laninamivir: 30 (13/17)	All groups between 4-12	≤18	NO	Intravenous with peramivir 10 mg/kg once daily or oral oseltamivir 2 mg/kg twice daily, ≧37.5 kg or oral oseltamivir 75 mg twice daily for 5 days or inhaled zanamivir 10 mg twice daily for 5 days or inhaled laninamivir 40 mg (≥10 years) or 20 mg (<10 years) once daily.	Time to virus clearance, time to alleviation of fever, time to alleviation of symptoms, adverse events.

**Table 5 medicina-56-00063-t005:** Summary of neuraminidase inhibitors used for influenza treatment in observational studies in the meta-analysis.

Author, Year	Influenza Virus Subtype	Patient Enrollment Criteria	Sample Number (Men/Women)	Age (Mean ± SD)	Age Group	Treatment Protocol	Outcome Measurement
**Observational studies**							
**Sakata H, 2011**	A B	Rapid test positive, influenza-like illness within the previous 48 h.	Peramivir: 30 (N/A) Oseltamivir: 30 (N/A)	Peramivir: 1.8 ± 4.9 Oseltamivir: 2.0 ± 3.9	≤18	Intravenous with peramivir 10 mg/kg once daily or oral oseltamivir 4 mg/kg daily.	Time to alleviation fever, adverse events.
**Hikita T et al., 2012**	A B	Rapid test positive, fever lasting for less than 48 h.	Peramivir: 63 (N/A) Oseltamivir: 124 (N/A) Zanamivir: 38 (N/A) Laninamivir: 14 (N/A)	Peramivir: 7.8 ± 42.4 Oseltamivir: 5.2 ± 34.2 Zanamivir: 10.5 ± 11.4 Laninamivir: 10.6 ± 5.8	≤18	Intravenous with peramivir 10 mg/kg once daily or oral oseltamivir 2 mg/kg twice daily for 5 days or inhaled zanamivir 10 mg twice daily for 5 days or inhaled laninamivir 40 mg (≥10 years) or 20 mg (<10 years) once daily.	Time to alleviation fever, adverse events.
**Shobugawa Y et al., 2012**	A(H3N2), A(H1N1) pdm09	Rapid test positive, fever ≥ 37.5 °C with respiratory symptoms, headache, arthralgia, or myalgia	Peramivir: 4 (3/1) Oseltamivir: 119 (60/59) Zanamivir: 124 (78/46) Laninamivir: 9 (3/6)	Peramivir: 8.8 ± 3.9 Oseltamivir: 4.9 ± 2.3 Zanamivir: 9.4 ± 2.5 Laninamivir: 10.2 ± 2.3	Mix	Intravenous with peramivir 300 or 600 mg with adult or 10 mg/kg for children once daily or oral oseltamivir 75 mg twice daily with adult or 2 mg/kg twice daily with children for 5 days or inhaled zanamivir 10 mg twice daily for 5 days or inhaled laninamivir 40 mg (≥10 years) or 20 mg (<10 years) once daily.	Time to alleviation fever.
**Takemoto Y et al., 2013**	A B	Rapid test positive.	Peramivir: 53 (32/21) Oseltamivir: 51 (26/25) Zanamivir: 39 (28/11) Laninamivir: 44 (25/19)	Peramivir: 34.8 ± 23.2 Oseltamivir: 19.0 ± 27.0 Zanamivir: 17.8 ± 18.6 Laninamivir: 26.3 ± 23.2	Mix	Intravenous with peramivir 300 mg with adult or 10 mg/kg for children once daily or oral oseltamivir 75 mg twice daily with adult or 2 mg/kg twice daily with children for 5 days or inhaled zanamivir 10 mg twice daily for 5 days or inhaled laninamivir 40 mg (adult) or 20 mg (children) once daily.	Time to alleviation fever.
**Sugaya N et al., 2015**	A(H3N2), A(H1N1) pdm09	Rapid test positive, fever >38 °C, upper respiratory symptoms such as cough or rhinorrhoea.	Peramivir: 17 (N/A) Oseltamivir: 163 (N/A) Laninamivir: 33 (N/A)	Peramivir: 7.6 ± 3.8 Oseltamivir: 6.3 ± 1.8 Laninamivir: 8.3 ± 2.0	≤18	Intravenous with peramivir 10 mg/kg once daily or oral oseltamivir weight-based dose twice daily for 5 days or inhaled laninamivir 20 mg once daily.	Viral shedding patterns, time to alleviation fever
**Yoshino Y et al., 2015**	Not mentioned	Rapid test positive, oral temperature ≥37.2 °C, influenza-like illness.	Peramivir: 23 (14/9) Oseltamivir: 9 (4/5)	Peramivir: 77.6 ± 14.4 Oseltamivir: 70.3 ± 13.8	>18	Intravenous with peramivir 300 mg once daily or oral oseltamivir 75 mg twice daily for 5 days.	Time to defervescence, survival rate, side effects
**Tochino Y et al., 2017**	A B	Rapid test positive, fever ≥ 37 °C influenza-like illness.	Peramivir: 10 (4/6) Oseltamivir: 108 (55/53) Zanamivir: 28 (14/14) Laninamivir: 95 (43/52)	Peramivir: 44.0 ± 53.2 Oseltamivir: 25.3 ± 343.3 Zanamivir: 15.8 ± 70.9 Laninamivir: 33.0 ± 273.5	Mix	Not mentioned	Time to alleviation fever, time to alleviation of symptoms, adverse events.

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
