# Peer review of "A Meta-Analysis Comparing the Efficacy and Safety of Peramivir with Other Neuraminidase Inhibitors for Influenza Treatment"

_medicina, 2020, doi:10.3390/medicina56020063_

Round 1
Reviewer 1 Report
A meta-analysis comparing the efficacy and safety of peramivir with other neuraminidase inhibitors for influenza treatment.
In this manuscript, Chen et al. analyze published studies to determine the difference in time to alleviation of symptoms of peramivir versus other NAIs as well as compare adverse effects between the antivirals. While they find that peramivir decreases the time of symptoms, the authors leave out a lot of information in their writing to the reader, as well as overlook what these numbers would mean to a person with an influenza infection.
General Comments:
What were the range times of alleviation of symptoms in these studies? The mean differences between 2 of the drugs only gives the reader the difference, instead of what that would me in actual life with influenza. This is especially important as the difference in time to alleviation of symptoms of peramivir vs. oseltamivir is less than half a day. The authors should also use this information to potentially highlight the almost 1 day difference of peramivir vs. non-oseltamivir NAIs
Section 2.3: Please describe the GRADE system in more detail for the readers and reviewers.
Figures 2-9: The authors do not indicate what the black shapes are in the figure legends (assuming a value that is a combination of all the studies listed), how it was calculated, and how the relative weight was determined of each study. Please add.
Line 191-192: Did Lee et al. determine the time to alleviation for peramivir and oseltamivir? If so is the difference similar to your findings?
Lines 205-208: Remove the first sentence of your paragraph. You can talk about how your study was different from the other, but only specify limitations of your study.
Lines 212-213: The authors talk about the possibility of vaccination status affecting time of alleviation. Please indicate why you did not account for this (if the papers did not specify and therefore the authors could not perform that analysis, etc.)
Specific/Writing Comments:
Line 14: change “with” to “compared to” for clarity (looking at safety of peramivir vs. other NAIs instead of in combination like it now reads).
Line 138 (Figure 2 legend). The “overall effect” is odd wording. This should be labeled Peramivir vs. all other NAIs
Lines 140-141 (Figure 3 legend): Please define OS/RCT
Figures 4-6: “with other neuraminidase inhibitors” can be removed to be concise (peramivir vs. laninamivir, etc.)
Author Response
Point 1. In this manuscript, Chen et al. analyze published studies to determine the difference in time to alleviation of symptoms of peramivir versus other NAIs as well as compare adverse effects between the antivirals. While they find that peramivir decreases the time of symptoms, the authors leave out a lot of information in their writing to the reader, as well as overlook what these numbers would mean to a person with an influenza infection.
Response 1.
Yes. We thank the reviewer for the insightful comments. All comments and our responses are listed in the following tables to each of the reviewers, respectively.
Point 2. What were the range times of alleviation of symptoms in these studies? The mean differences between 2 of the drugs only gives the reader the difference, instead of what that would me in actual life with influenza. This is especially important as the difference in time to alleviation of symptoms of peramivir vs. oseltamivir is less than half a day. The authors should also use this information to potentially highlight the almost 1 day difference of peramivir vs. non-oseltamivir NAIs
Response 2.
Yes. We thank the reviewer for the insightful comments. The range of time of alleviation of symptoms in t hese studies: peramivir group was 7.2 to 204.4 hours and nonperamivir group was 5.7 to 96.5 hours.
We discussed the importance of the difference in time to alleviation between peramivir and other NAIs on lines 208-211: This meta-analysis showed peramivir reduced the time to alleviation of symptoms 12 to 24 hours compared to other NAIs, this data potentially mean that peramivir can shorten hospitalization stays, decrease medical costs, avoid complications and return to normal life quickly.
Point 3.Section 2.3: Please describe the GRADE system in more detail for the readers and reviewers.
Response 3. General to say, the GRADE system is to evaluate each study by study design, risk of bias, inconsistency, indirectness and imprecision, then you can rank the quality of each study. The quality has four-level, high, moderate, low and very low. For detail please see “GRADE: an emerging consensus on rating quality of evidence and strength of recommendations. BMJ 2008; 336”.
Point 4. Figures 2-9: The authors do not indicate what the black shapes are in the figure legends (assuming a value that is a combination of all the studies listed), how it was calculated, and how the relative weight was determined of each study. Please add.
Response 4.
We thank the reviewer for the insightful comments. Black shapes indicate overall summary. The overall summary was calculated by CMA v3 software. How the relative weight was determined of each meta-analysis software were the same, for detail please see “Borenstein M, et al. A basic introduction to fixed-effect and random-effects models for meta-analysis. Res Synth Methods. 2010 1(2):97-111”
Point 5. Line 191-192: Did Lee et al. determine the time to alleviation for peramivir and oseltamivir? If so is the difference similar to your findings?
Response 5. Yes, Lee et al. determined the time to alleviation for peramivir and oseltamivir. The result was similar to our finding, but we included more recent studies.
Point 6. Lines 205-208: Remove the first sentence of your paragraph. You can talk about how your study was different from the other, but only specify limitations of your study.
Response 6.
We thank the reviewer for the insightful comments. We reword the first sentence“This study has some differences from Lee et al.”.
Point 7. Lines 212-213: The authors talk about the possibility of vaccination status affecting time of alleviation. Please indicate why you did not account for this (if the papers did not specify and therefore the authors could not perform that analysis, etc.)
Response 7.
We thank the reviewer for the insightful comments. We remove the vaccination status affecting time of alleviation.
Point 8.Line 14: change “with” to “compared to” for clarity (looking at safety of peramivir vs. other NAIs instead of in combination like it now reads).
Response 8.
Yes, we thank the reviewer for the insightful comments. we modify text “This meta-analysis compared the efficacy and safety of peramivir compared to other neuraminidase inhibitors (NAIs)”.
Point 9. Line 138 (Figure 2 legend). The “overall effect” is odd wording. This should be labeled Peramivir vs. all other NAIs
Response 9.
Yes, we thank the reviewer for the insightful comments. We modify text “Time to alleviation of symptoms of peramivir compared with other neuraminidase inhibitors: Peramivir vs. all other NAIs”. (hours)
Point 10. Lines 140-141 (Figure 3 legend): Please define OS/RCT
Response 10.
Yes, Yes, we completely agree with the reviewer’s comment. We add OS: observational study, RCT: randomized controlled trial on Figure 3 legend.
Point 11. Figures 4-6: “with other neuraminidase inhibitors” can be removed to be concise (peramivir vs. laninamivir, etc.)
Response 11
Yes, Yes, we completely agree with the reviewer’s comment. We modify text
Figure 4: Time to alleviation of symptoms of peramivir vs. oseltamivir. (hours)
Figure 5: Time to alleviation of symptoms of peramivir vs. zanamivir. (hours)
Figure 6: Time to alleviation of symptoms of peramivir vs. laninamivir. (hours)
Reviewer 2 Report
(1) The dose and frequency of oseltamivir are constant while the dose and frequency of peramivir vary from study to study. This is likely to have had a significant impact on the results. It will be good to do a subgroup analysis if possible and, if not possible, describe the limitations in more detail.
(2) Even considering the contents of limitation, the effects on influenza A and B need to be analyzed separately. If it is possible to carry out the analysis, it is recommended to perform the subgroup analysis and present it in consideration of the limitations.
(3) For adults, it is necessary to analyze elderly and non-elderly adults based on 65 years of age.
(4) Although the time to fever loss is the most important factor in clinical efficacy analysis, another point of concern with peramivir is the effect in severe patients. This study does not focus on this area, and the analysis of selected studies does not seem feasible. I recommend to mention this in the discussion.
Author Response
Point 1. The dose and frequency of oseltamivir are constant while the dose and frequency of peramivir vary from study to study. This is likely to have had a significant impact on the results. It will be good to do a subgroup analysis if possible and, if not possible, describe the limitations in more detail.
Response 1.
Yes. We thank the reviewer for the insightful comments. Due to inconsistency of study design among these studies, we did not investigate the dosage, duration of NAI treatment in this meta-analysis, it hindered carrying out subgroup analysis of elderly and non-elderly adults and influenza subtype as well.
Point 2. Even considering the contents of limitation, the effects on influenza A and B need to be analyzed separately. If it is possible to carry out the analysis, it is recommended to perform the subgroup analysis and present it in consideration of the limitations.
Response 2.
Yes. We thank the reviewer for the insightful comments. Due to inconsistency of study design among these studies, we did not investigate the dosage, duration of NAI treatment in this meta-analysis, it hindered carrying out subgroup analysis of elderly and non-elderly adults and influenza subtype as well.
Point 3. For adults, it is necessary to analyze elderly and non-elderly adults based on 65 years of age.
Response 3.
Yes. We thank the reviewer for the insightful comments. Due to inconsistency of study design among these studies, we did not investigate the dosage, duration of NAI treatment in this meta-analysis, it hindered carrying out subgroup analysis of elderly and non-elderly adults and influenza subtype as well.
Point 4. Although the time to fever loss is the most important factor in clinical efficacy analysis, another point of concern with peramivir is the effect in severe patients. This study does not focus on this area, and the analysis of selected studies does not seem feasible. I recommend to mention this in the discussion
Response 4.Yes. We thank the reviewer for the insightful comments. This study did not focus on severe patients because studies investigating severe patients on influenza treatment by NAIs are limited.
Reviewer 3 Report
Summary & Overall Assessment
This systematic review & meta-analysis compared the efficacy and safety of peramivir with other neuraminidase inhibitors for the treatment of influenza. Overall, the manuscript is well-written, and the analyses are sound. However, my biggest concern is regarding the clinical relevance of the primary outcome – i.e. time to alleviation of symptoms, with differences measured in hours. Given that peramivir is administered intravenously, it is unlikely that the drug will be used outside of a hospital setting, and thus, examination of outcomes that are more relevant to severe influenza may have greater clinical utility. At the minimum, the authors should discuss the limitations of their primary outcome and place the magnitude of their results in the context of clinical practice, rather than focus on their significance. In addition to this major critique, the reviewer has the following minor comments for the authors’ consideration.
Background
Please add information re: the recent meta-analysis comparing peramivir to oseltamivir (Lee et al., reference 21) & clearly identify the knowledge gap that remains that your study aims to address
Methods
Please indicate whether your inclusion criteria required the administration of peramivir/other NAIs for influenza treatment only (i.e. not prophylaxis studies – particularly relevant for AE/SAE studies) Please indicate whether you had multiple reviewers conducting the data extraction & quality assessments Please specify the unit for the measurement of time to alleviation of symptoms (hours)
Results
Your flow chart doesn’t seem to match the narrative for the 1 study excluded for crossover treatment (in the flowchart, this article was described as excluded for “lack of data”) - please revise. Alternatively, if these do not refer to the same article, please clarify this in your article. Please include the unit of measure (hours) for time to alleviation of symptoms in Figures 2-9 Page 9, lines 128-129 – please clarify that you mean sub-analyses comparing specific NAIs Lines 133-135, please reword the following passages for clarity: “exhibited a significantly favor peramivir”, “exhibited favor peramivir, but nonsignificant effects” Section 3.3 – with so few studies, can you please specify the NAI comparators for AE & SAE analyses (either in text or figure)
Discussion
As indicated above, please discuss the clinical utility of your outcome, particularly in the context of peramivir administration in predominantly severe/hospitalized influenza cases. Please ensure that you also discuss the magnitude of your efficacy findings, as well as their significance.
Author Response
Point 1. This systematic review & meta-analysis compared the efficacy and safety of peramivir with other neuraminidase inhibitors for the treatment of influenza. Overall, the manuscript is well-written, and the analyses are sound. However, my biggest concern is regarding the clinical relevance of the primary outcome – i.e. time to alleviation of symptoms, with differences measured in hours. Given that peramivir is administered intravenously, it is unlikely that the drug will be used outside of a hospital setting, and thus, examination of outcomes that are more relevant to severe influenza may have greater clinical utility. At the minimum, the authors should discuss the limitations of their primary outcome and place the magnitude of their results in the context of clinical practice, rather than focus on their significance. In addition to this major critique, the reviewer has the following minor comments for the authors’ consideration.
Response 1.
Yes. We agree with the reviewer’s concern. All comments and our responses are listed in the following tables to each of the reviewers, respectively.
Point 2. Please add information re: the recent meta-analysis comparing peramivir to oseltamivir (Lee et al., reference 21) & clearly identify the knowledge gap that remains that your study aims to address
Response 2.
Yes, we completely agree with the reviewer’s comment. We add some information on lines 47-48 (But recent meta-analysis [21] comparing peramivir to oseltamivir only determined the efficacy on adult group.) and lines 51-52 (This meta-analysis compared the clinical efficacy of peramivir with the three other NAIs with respect to time to alleviation of symptoms or defervescence on groups of more than 18 years and less than or equal to 18 years)
Point 3. Please indicate whether your inclusion criteria required the administration of peramivir/other NAIs for influenza treatment only (i.e. not prophylaxis studies – particularly relevant for AE/SAE studies) Please indicate whether you had multiple reviewers conducting the data extraction & quality assessments Please specify the unit for the measurement of time to alleviation of symptoms (hours)
Response 3.
Yes, we completely agree with the reviewer’s comment.
The inclusion criteria were rapid test positive and administration of NAIs for influenza treatment only. Please see lines 61-63. (Only articles with patients were rapid test positive comparing intravenous peramivir with at least one other NAI were included.)
Lines 72 data extraction by two reviewers (T.-S.W. and S.-K.W.) and lines 88-90 two reviewers (T.-S.W. and C.-C.L.) evaluated all quality of articles to avoid bias. When they disagreed on the quality of an article, a third author (S.-K.W.) judged the inclusion of the article.
Lines 76 using a random-effect model to analyze time to alleviation of symptoms (hours)
Point 4. Your flow chart doesn’t seem to match the narrative for the 1 study excluded for crossover treatment (in the flowchart, this article was described as excluded for “lack of data”) - please revise. Alternatively, if these do not refer to the same article, please clarify this in your article. Please include the unit of measure (hours) for time to alleviation of symptoms in Figures 2-9 Page 9, lines 128-129 – please clarify that you mean sub-analyses comparing specific NAIs Lines 133-135, please reword the following passages for clarity: “exhibited a significantly favor peramivir”, “exhibited favor peramivir, but nonsignificant effects” Section 3.3 – with so few studies, can you please specify the NAI comparators for AE & SAE analyses (either in text or figure)
Response 4.
Yes, we completely agree with the reviewer’s comment. Flow chart did not match text at lines 98-99 because we missed reference 32, we very thank the reviewer noted this error, therefore lines 98-99 we added “and the other one was excluded because lack of data [32]”.
The unit of measure (hours) for time to alleviation of symptoms in figures 2-7 was added as below. Figure 8-9 risk ratio are not suitable for unit of measure (hours).
Figure 2: Time to alleviation of symptoms of peramivir compared with other neuraminidase inhibitors: Peramivir vs. all other NAIs. (hours)
Figure 3: Time to alleviation of symptoms of peramivir compared with other neuraminidase inhibitors: subgroup analysis by study design. (hours) (OS: observational study, RCT: randomized controlled trial)
Figure 4: Time to alleviation of symptoms of peramivir vs. oseltamivir. (hours)
Figure 5: Time to alleviation of symptoms of peramivir vs. zanamivir. (hours)
Figure 6: Time to alleviation of symptoms of peramivir vs. laninamivir. (hours)
Figure 7: Time to alleviation of symptoms of peramivir compared with other neuraminidase inhibitors: More than 18 years and Less than or equal to 18 years. (hours)
Lines 135-136 was modified as “the peramivir group exhibited significantly shorter time to alleviation of symptoms than the oseltamivir group MD = −11.338 hours, 95% CI: −19.475 to −3.200, p = 0.006, Figure 4; peramivir vs zanamivir: MD = −20.846 hours, 95% CI: −31.333 to −10.359, p < 0.05, Figure 5; peramivir vs laninamivir: MD = −21.571 hours, 95% CI: −29.656 to − 13.486, p < 0.05, Figure 6”
Line 140-142 was modified as “exhibited significant favor towards peramivir”, “exhibited favor towards peramivir, but without statistical significance”
In section 3.3, the nonperamivir was mentioned in these studies was oseltamivir. We modified the text and figures as below.
The risk of AE in the peramivir group was similar to that in the oseltamivir group (risk ratio [RR] = 1.023, 95% CI: 0.717 to 1.460, p = 0.900; Figure 8). The risk of SAE was similar between the peramivir and oseltamivir groups (RR = 1.068, 95% CI: 0.702 to 1.625, p = 0.759; Figure 9).
Figure 8. Risk of adverse events for peramivir compared with oseltamivir: overall risk of adverse event.
Figure 9. Risk of adverse events for peramivir compared with oseltamivir: overall risk of serious adverse event.
Point 5. As indicated above, please discuss the clinical utility of your outcome, particularly in the context of peramivir administration in predominantly severe/hospitalized influenza cases. Please ensure that you also discuss the magnitude of your efficacy findings, as well as their significance.
Response 5.
Yes. We thank the reviewer for the insightful comments. We discussed the clinical utility on lines 208-211: This meta-analysis showed peramivir reduced the time to alleviation of symptoms 12 to 24 hours compared to other NAIs, this data potentially mean that peramivir can shorten hospitalization stays, decrease medical costs, avoid complications and return to normal life quickly.